# Preterm Perinatal Hypoxia-Ischemia Does not Affect Somatosensory Evoked Potentials in Adult Rats

**DOI:** 10.3390/diagnostics9030123

**Published:** 2019-09-18

**Authors:** Melinda Barkhuizen, Johan S.H. Vles, Ralph van Mechelen, Marijne Vermeer, Boris W. Kramer, Peter Chedraui, Paul Bergs, Vivianne H.J.M. van Kranen-Mastenbroek, Antonio W.D. Gavilanes

**Affiliations:** 1Department of Pediatrics, Maastricht University Medical Centre (MUMC), 6229HX, Maastricht, The Netherlands; m.barkhuizen@maastrichtuniversity.nl (M.B.); r.vanmechelen@maastrichtuniversity.nl (R.v.M.); marijne.vermeer@gmail.com (M.V.); b.kramer@mumc.nl (B.W.K.); 2Department of Translational Neuroscience, School for Mental Health and Neuroscience (MHeNs), Maastricht University, 6229 HX, Maastricht, The Netherlands; jsh.vles@mumc.nl; 3DST/NWU Preclinical Drug Development Platform, North-West University, Potchefstroom 2531, South Africa; 4Child Neurology, Maastricht University Medical Centre, 6229 HX, Maastricht, The Netherlands; 5Instituto de Investigación e Innovación de Salud Integral, Facultad de Ciencias Médicas, Universidad Católica de Santiago de Guayaquil, Guayaquil 090615, Ecuador; peterchedraui@yahoo.com; 6Clinical Neurophysiology, Maastricht University Medical Centre, 6229 HX Maastricht, The Netherlands; p.bergs@mumc.nl (P.B.); v.kranen.mastenbroek@mumc.nl (V.H.J.M.v.K.-M.)

**Keywords:** somatosensory evoked potential, preterm brain, rat model, cerebral palsy, hypoxia-ischemia

## Abstract

Somatosensory evoked potentials (SSEPs) are a valuable tool to assess functional integrity of the somatosensory pathways and for the prediction of sensorimotor outcome in perinatal injuries, such as perinatal hypoxia-ischemia (HI). In the present research, we studied the translational potential of SSEPs together with sensory function in the male adult rat with perinatal HI compared to the male healthy adult rat. Both somatosensory response and evoked potential were measured at 10-11 months after global perinatal HI. Clear evoked potentials were obtained, but there were no group differences in the amplitude or latency of the evoked potentials of the preceding sensory response. The bilateral tactile stimulation test was also normal in both groups. This lack of effect may be ascribed to the late age-of-testing and functional recovery of the rats.

## 1. Introduction

Somatosensory evoked potentials (SSEPs) are a valuable tool to assess functional integrity of the somatosensory pathways of the peripheral and central nervous systems [1]. SSEPs are widely used for the prediction of motor outcomes in perinatal injuries, where the long-term effects of the injury are difficult to ascertain early in life [2]. Hypoxic-ischemic encephalopathy (HIE) is a common injury in neonates, with a global occurrence of 8.5 infants per 1000 live births in 2010 [3]. Infants with HIE, especially preterm infants, are considered to be at risk for developmental disorders [4]. They present a heterogeneous clinical picture, varying by individual, and ranging from major disorders like cerebral palsy (CP) to less severe impairments like developmental coordination disorder (DCD). DCD is commonly associated with other developmental comorbidities, including attention deficit/hyperactivity disorder (ADHD) and learning disabilities, and could be related to an impaired sensory process.

Several studies have shown that SSEPs, in combination with other electrophysiological tools and neuroimaging, are useful in the diagnosis of encephalopathy after HIE and could improve the prediction of infants that will have poor neurodevelopment outcomes two years after perinatal HIE [2,5,6,7]. However, the predictive value of SSEPs decreases with longer follow-up periods. Neonatal SSEP latencies do not correlate with neurodevelopmental outcomes at school-going age in infants with mild HIE. The long-term prognostic value of neonatal SSEPs in this patient group is unclear [1]. It is also unknown whether the neonatal SSEP abnormalities persist in adults with HIE. On the other hand, SSEPs are suited to evaluate patients suffering from more severe forms of disability such as cerebral palsy (CP), in whom neurological deficits might reflect disruption of motor as well as sensory connections [8,9], suggesting a neural network disorder. Furthermore, both motor and behavioral skills involve the process of receiving a sensory input. In this context, motor-behavioral deficits could be treated by sensory-based therapies. Indeed, this may be acceptable as one component of a comprehensive treatment plan in the management of children with developmental, behavioral, and motor disorders of different etiology [10]. 

SSEP is a relatively simple, objective, and reproducible diagnostic procedure that assesses the effect of the somatosensory input in the peripheral and central neuronal networks. In the present experiments, we used a perinatal rat model of hypoxic-ischemia (HI) injury by submersion [11]. Previous studies using this model of global perinatal preterm HI demonstrated that severe HI (submersion lasting for 19–20 min) decreased locomotor activity in the adult rat, while milder HI insults might increase locomotion [12,13]. In this respect, our experimental group is a cohort of male adult rats that recovered from moderate global perinatal HI and showed cognitive (memory) as well as motor abnormalities (hyperactivity) within the spectrum of ADHD at the age of 6–8 months [14]. To test whether this cohort had a problem in coordination (DCD), we evaluated the sensory functioning using a bilateral tactile stimulation test and we recorded cortical SSEPs after unilateral tibial nerve stimulation at 11 months of age [14].

The present study reports the outcomes of the SSEPs and behavioral tasks of sensory functioning of male adult rats subjected to moderate perinatal HI. We hypothesized that long-term sensory deficits were relevant to the combined sensory and motor deficits previously demonstrated in this cohort of male rats after perinatal HI.

## 2. Materials and Methods

### 2.1. Rodents and Ethics Approval

The experiments were performed in Sprague-Dawley rats from Charles River (Leiden, The Netherlands). The rats were housed at the Central Animal Experimentation Facilities of the Maastricht University, The Netherlands. Experimental female rats were synchronized with luteinizing hormone-releasing hormone (Cat. L4513, Sigma-Aldrich, The Netherlands), and time-mated between 15:00 and 07:00 the following morning. All experiments were conducted under ethical approval from the Dutch Central Committee for Animal Testing according to the guidelines of the EU directive 2010/63/EU (approval code: AVD107002016540, approved on 16 July 2016).

### 2.2. Perinatal Hypoxia-Ischemia (HI) Procedure

In the afternoon of embryonic day 21 (expected delivery on E21–E22), a pregnant female rat was euthanized by rapid decapitation and the uterine horns, still containing the pups, rapidly removed and submerged in saline solution at 37 °C for 16–18 min according to previously-described methodology [12,13]. After submersion, the pups were delivered, manually stimulated to breathe, and placed in a closed pediatric incubator to recover for an hour (HI group). The control Cesarean-section rats (C-section) were delivered from the same litters without submersion and placed in the closed pediatric incubator to recover with their HI littermates. After recovery, the mixed litter of HI and C-section pups was placed with a foster mother that had given birth the day before to minimize the impact of maternal care differences on outcomes. The mortality rate for the HI procedure was 39%. 

### 2.3. Housing

After weaning, the rats were fed a standard laboratory diet and housed in pairs in individually ventilated cages, up to 8 months of age, when they were housed 4 per cage in larger filter top cages. The rats were housed on a reverse day night cycle (lights off at 7:00/lights on at 19:00). 

### 2.4. Adhesive Removal (‘Sticker Test’)

The adhesive removal test was performed at 10 months of age. Somatosensory response was assessed with a bilateral tactile stimulation test adapted from a method previously reported [15,16]. For the sticker test, rats were placed in an empty cage with videotaping from the sides. One investigator restrained the rat, whilst another investigator placed two brightly colored circular adhesive labels (1.27 cm diameter, Avery office products, Houten, The Netherlands) on the dorsum of both forepaws of the rat. Adult rats normally touch and remove the stickers with their teeth. The time from placing the rat in the arena to the initial purposeful sticker contact (‘noticed’) and to removal from both paws (‘removed’) was recorded and the latency time from initial contact to removal was calculated. The time limit for adhesive removal was 180 s. Group sizes for this test were C-section (*n* = 12) and HI (*n* = 11).

### 2.5. Sedation

For the SSEPs, the rats were placed under anesthesia with an induction of midazolam (0.5 mg/kg), followed by a mixture of ketamine (0.75 mg/kg), medetomidine (0.06 mg/kg) and atropine (0.04 mg/kg) in saline (KMA). Sedation was maintained during the procedure with an intraperitoneal infusion of the KMA mixture.

### 2.6. Somatosensory Evoked Potentials (SSEPs)

The SSEP was conducted according to methodology previously described by Zhang, et al. [17] at the age of 11 months. A Nicolet Viking IV P™ (Nicolet Biomedical, Madison, WI, USA) was utilized to record and analyze the waveform of SSEPs. Twelve-millimeter long monopolar needle electrodes with attached lead wire were used for active, reference, and ground electrodes. A sedated rat was placed on a board and the needle electrodes were carefully inserted into the scalp and placed on the surface of the skull, parallel with the long axis of the rat body. The active electrode was located on the midline of the skull and crossed the point of bregma for right hemisphere recording.

The reference electrode was placed on the skull surface over the olfactory bulb and the ground electrode was placed in the shoulder. Left tibial nerve stimulation was performed with 0.2 ms pulses at 1.7 Hz (filter set to 2–3000 Hz) with an intensity of 1–3 mA, depending on the twitch response, for an average of 500 stimuli. We studied the cortical SSEPs P_1_ and N_2_ amplitudes and latencies, which were registered by a child neurologist (JSHV) and reviewed together with a clinical neurophysiologist (VHvKM) and a clinical neurophysiology technician (PB). 

We recorded 6 male rats per group, for the following groups: C-section (body weights: 601 ± 36 g) and HI (body weights: 574 ± 33 g).

### 2.7. Statistics

Stata 10 (Statacorp, TX, USA) was used to assess group differences by means of the non-parametric Kruskal-Wallis H-test followed by post-hoc Dunn’s test of equality. Regarding SSEPs, the averaged value between two runs was used. *P* values <0.05 were considered as statistically significant. Figures regarding average ± standard error means were constructed with GraphPad Prism 6 (GraphPad Software Inc., La Jolla, CA, USA). Outliers were not considered in the analysis.

## 3. Results

### 3.1. Adhesive Removal Test (‘Sticker Test’)

The results of the sticker test are shown in Figure 1. There were no significant differences between groups regarding the time the rats took to notice the sticker on their paws (*p* = 0.943), the time to remove it (*p* = 0.546), or the time from notice to removal (*p* = 0.244).

### 3.2. Evoked Potentials

Reproducible SSEPs were obtained in all rats. After exclusion of one outlier, our group sizes were: C-section (*n* = 5) and HI (*n* = 6). There were no significant differences for the amplitude of the P_1_ peak (*p* = 0.078), the N_2_ peak (*p* = 0.891), or the absolute difference in the amplitude of the P_1_-N_2_ peaks (*p* = 0.469). There were also no significant group effects on the latencies of these peaks P_1_ (*p* = 0.667), N_2_ (*p* = 0.558) (Figure 2 and Figure 3). Exclusion of the outlier did not statistically change the results.

## 4. Discussion

In human newborns, particularly those born preterm, HI is an important cause of long-term neurological disabilities, such as learning and memory deficits and sensorimotor and motor functioning deficits like CP [18,19]. The present study investigated the long-term effects of moderate HI on somatosensory functioning in male survivors using an established rat model of preterm HI.

Recording of SSEPs is a quantitative method for evaluation of both the central and peripheral nervous system. Peripheral sensory information reaches the parietal-occipital cortex by thalamocortical pathways e.g., corona radiata and internal capsule. The parietal cortex connects to neocortical areas including premotor, prefrontal areas as well as to the cerebellum through pontine nuclei. Merging this information with the basal ganglia, pre-Rolandic motor areas control motor activity through the descending corticospinal tracts [20]. SSEPs are suited to evaluate children and adult patients suffering from CP [9]. Through the findings of neuroimaging studies, a correlation has been reported in CP patients between disrupted descending corticospinal pathways and disrupted thalamocortical pathways connecting to the sensory cortex [20]. The role of SSEPs in children and adults with milder deficits after HI is unclear. Identifiable sensory processing and motor interaction during early childhood, e.g., reaching and touching objects using hands and mouth, provides a critical foundation for normal growth, development, and learning. This sensorimotor integration explains why sensory impairments affect motor recovery, and why sensory based strategies might promote motor and behavioral recovery. 

The seminal work reported by de Louw, et al. (2002) on the short-term effect of severe perinatal HI on spinal cord apoptosis using this perinatal model of submersion showed an increased lumbar grey and white-matter apoptosis [21]. These changes may contribute to the permanent motor deficits, which are the main neurological manifestations of brain injury in the premature infant. We investigated whether preterm HI had a long-term effect on expected sensorimotor deficits.

In the adhesive removal test, we did not observe a statistically significant effect of HI. We used the time for removal from both paws, since we caused a global insult which was expected to affect both hemispheres equally. In adult rats with unilateral lesions, a bias to touching the non-affected forepaw first has been reported [15]. A previous study of neonatal unilateral hypoxic-ischemic injury also did not find significant group differences with the adhesive removal among five-week-old rats with a modest brain injury [16].

Tibial SSEPs evaluate the somatosensory pathway including dorsal columns of the rat’s spinal cord and the contralateral parietal cortex. Our study found that the perinatal HI insult did not influence the SSEPs. A previous study with SSEPs after neonatal stroke in the rat, found profound unilateral changes in SSEPs one week after the insult, but recovery of the deficits 3 weeks postnatally. This may indicate large-scale plasticity of the somatosensory networks even after a unilateral neonatal injury [22]. It is thus possible that sufficient recovery occurred at 11 months after the global asphyxic insult and ameliorated changes observed in evoked potentials.

SSEPs were performed unilaterally (left tibialis nerve stimulation) assuming symmetrical and global central nervous system impact inherent to the global perinatal hypoxic-ischemic insult. This statement has been corroborated by the symmetrical adhesive removal test, which complements the SSEP data. In summary, we did not find any permanent significant sensory behavioral or electrophysiological changes in rat adulthood explaining the hyperactivity disorder and recognition memory deficit shown by this rat cohort. However, these results raise the question whether the somatosensory functioning of our rats has sufficiently recovered over time, or whether the SSEPs was not sensitive enough to detect deficits in adult rats.

There are several methodological and study limitations that could explain why we failed to detect preclinical SSEPs:(a)Due to our cross-sectional design, we might have missed SSEP changes in early postnatal age and adolescence.(b)Midazolam, ketamine and medetomidine were used as systemic anesthetics during SSEPs recordings. Accordingly, this anesthetic combination may have influenced SSEPs results; however, both experimental groups received the same anesthetic regime [23]. In humans, SSEPs are known to be less sensitive to the injectable anesthetics that we used, than to inhalation anesthetics [24].(c)The lumbar spinal cord apoptosis observed in severe HI models (19–20 min of submersion) may not necessarily be the postnatal hallmark of the moderate HI model (16–18 min of submersion) used in this experiment [21].(d)Evidence supports more neurological impairments and higher mortality for male preterm infants. In line with the human data, behavioral studies have consistently shown that the male sex is associated with an increased risk of long-term motor deficits in the rat preterm HIE [11]. Therefore, only male rats were included in this study.(e)Our global HI insult was conducted around the time of birth, when the rodent brain development is comparable to that of human infants born at a very low gestational age [11]. Studying SSEPs in larger animal models with brain development comparable to humans, such as sheep, may be more predictive of the human clinical situation [25]. However, larger animals are less suitable for chronic, long term studies.

Despite the aforementioned limitations, the present study found that, in the rat, moderate global HI did not influence the sensorimotor measurements assessed at 11 months. The moderate severity of the insult and the neuroplasticity observed early in life may explain the lack of a measurable effect on the adult survivors. More research is warranted to further confirm our results. 


**Highlights:**
SSEPs are used to clinically assess the integrity of peripheral and central somatosensory pathways after perinatal HI.Childhood SSEPs are often used to predict long-term outcomes, but it is unknown if sensory pathway deficits persist into adulthood.We showed the electrophysiological and behavioral integrity of the somatosensory pathways in adult rats subjected to moderate perinatal HI.


## Figures and Tables

**Figure 1 diagnostics-09-00123-f001:**
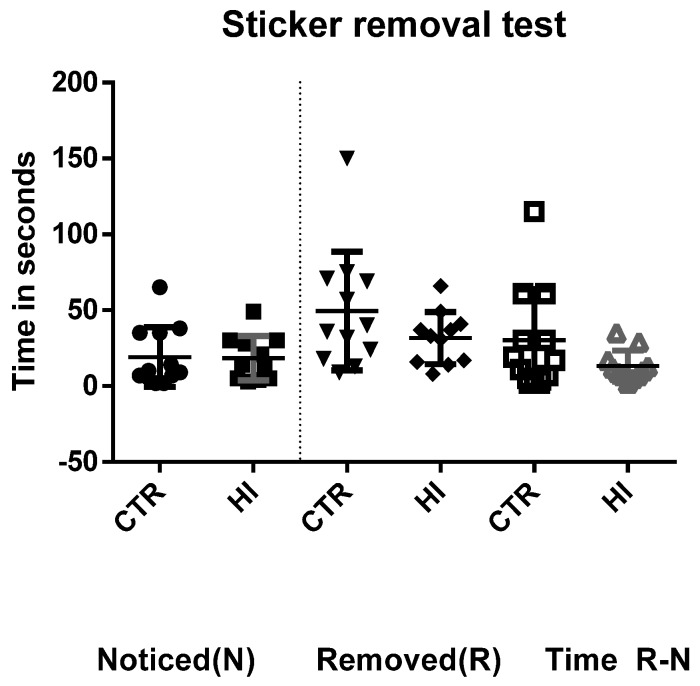
The adhesive removal test showing the latency to notice (N) and remove the stickers from the front paws (R), as well as the time elapsed between notice and removal (R-N). CTR: control C-section group, HI: hypoxia-ischemia group.

**Figure 2 diagnostics-09-00123-f002:**
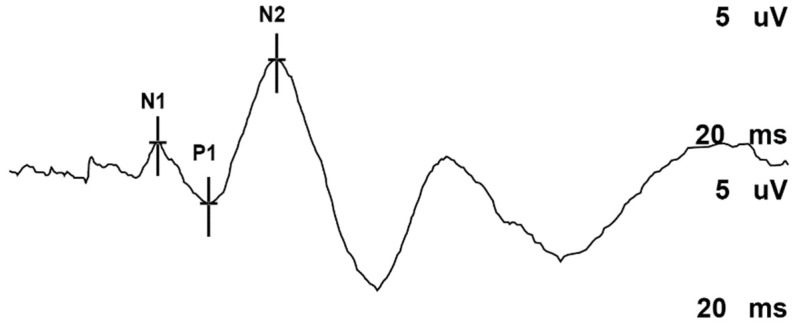
Representative example of tibial somatosensory evoked potentials (SSEPs) recording in adult rats. N_1_, P_1_ and N_2_ cortical latency and amplitude.

**Figure 3 diagnostics-09-00123-f003:**
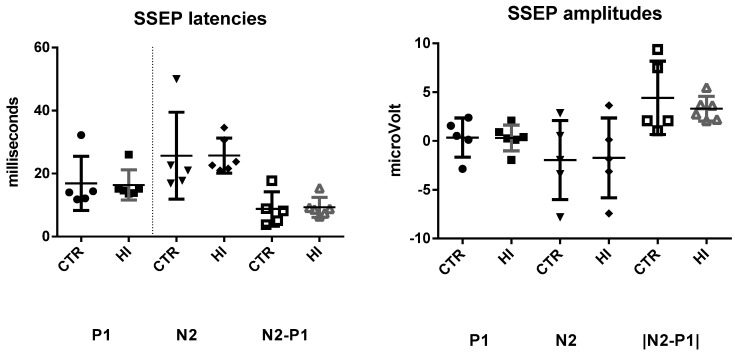
The latencies and amplitudes of the SSEP test. Each data point represents a single male animal; horizontal lines represent regional medians with the standard error medians. P1 latencies, N2 latencies, P1 amplitude, N2 amplitude, Absolute differences in amplitude. CTR: control C-section group, HI: hypoxia-ischemia group.

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
