# Peer review of "Preterm Perinatal Hypoxia-Ischemia Does not Affect Somatosensory Evoked Potentials in Adult Rats"

_diagnostics, 2019, doi:10.3390/diagnostics9030123_

Round 1

Reviewer 1 Report

1. Justify the reason for choosing 11 months age for the rats. 

2. The entire manuscript is based on results from SSEP analysis of HI impacted rats at the age of 11 months, but the authors have concluded that SSEP's can be used for predicting long term somatosensory outcomes immediately after HI. The results and the highlights of the manuscript are in contrary. 

3. Is SSEP incapable of detecting changes after an injury? HI injury does have an effect on the somatosensory circuit inspite of when the assessment is performed. Why is SSEP unable to detect the damage? 

The authors have to show data from SSEP experiments immediately after HI and that can be a selling point for the manuscript. 

Author Response

We would like to thank you for your time and the helpful feedback. We have proceeded to make a R1 version of our document making changes as suggested by the reviewers. Cut downs will not appear in the new R1 version. However, adjusted new text has been inserted as colored RED TEXT.

We thank the reviewers for their input and hope this revised version (R1) be accepted for publication. Of course, we look forward to further discussions on this manuscript.

Reviewer 1: 1 Justify the reason for choosing 11 months age for the rats. 

Reply: Thank you for this point. We have not been explicit enough, why we have performed the experiments at this age. We aimed to explain the observed cognitive and locomotion impairments found in this cohort of male rats in the months preceding the SSEP measurements [14]. At 11 months of age, the rats were approximately at the same development stage as a 28-year old human [15].  

We added: Lines 34-40. Infants with HIE, specially preterm infants, are considered to be at risk for developmental disorders [4]. They present a heterogeneous clinical picture, vary by individual and ranging from major disorders like cerebral palsy (CP) to less severe impairments like developmental coordination disorder (DCD). DCD is commonly associated with other developmental comorbidities, including attention deficit/hyperactivity disorder (ADHD) and learning disabilities, and could be related to an impaired sensory process.

Lines 62-71. In this respect, our experimental group is a cohort of male adult rats that recovered from moderate global perinatal HI, that has shown cognitive (memory) as well as motor abnormalities (hyperactivity) within the spectrum of ADHD at the age of 6-8 months [14]. To test whether this cohort has a problem in coordination (DCD), we evaluated the sensory functioning using a bilateral tactile stimulation test and we recorded cortical SSEPs after unilateral tibial nerve stimulation at 11 months of age [14].

The present study reports the outcomes of the SSEPs and behavioral tasks of sensory functioning of male adult rats subjected to moderate perinatal HI. We hypothesized that long-term sensory deficits were relevant to the combined sensory and motor deficits previously demonstrated in this cohort of male rats after perinatal HI.

Reviewer 1: 2. The entire manuscript is based on results from SSEP analysis of HI impacted rats at the age of 11 months, but the authors have concluded that SSEP's can be used for predicting long term somatosensory outcomes immediately after HI. The results and the highlights of the manuscript are in contrary. 

Reply: Thank you. We changed our second highlight (lines 232-233) to:

Childhood SSEPs are often used to predict long-term outcomes, but it is unknown if sensory pathway deficits persist into adulthood.

And we changed the introduction (lines 42-51): Several studies have shown that SSEPs, in combination with other electrophysiological tools and neuroimaging, are useful in the diagnosis of encephalopathy after HIE and could improve the prediction of infants that will have poor neurodevelopment outcomes 2 years after perinatal HIE [2,5-7]. However, the predictive value of SSEPs decrease with longer follow-up periods. Neonatal SSEP latencies do not correlate with neurodevelopmental outcomes at school-going age in infants with mild HIE. The long-term prognostic value of neonatal SSEPs in this patient group is unclear [1]. It is also unknown whether the neonatal SSEP abnormalities persist in adults with HIE. On the other hand, SSEPs are suited to evaluate patients suffering from more severe forms of disability, such as cerebral palsy (CP) in whom neurological deficits might reflect disruption of motor as well as sensory connections [8,9], suggesting a neural network disorder.

Reviewer 1: 3. Is SSEP incapable of detecting changes after an injury? HI injury does have an effect on the somatosensory circuit inspite of when the assessment is performed. Why is SSEP unable to detect the damage? 

Reply: The SSEPs technique should be capable of detecting somatosensory changes. We included our suspicions on why we did not observe changes in the discussion:

There are several methodological and study limitations that could explain why we failed to detect preclinical SSEPs:

(a) Due to our cross-sectional design, we might have missed SSEP changes in early postnatal age and adolescence.

(b) Midazolam, ketamine and medetomidine were used as systemic anaesthetics during SSEPs recordings. Accordingly, this anesthetic combination may have influenced SSEPs results; however both experimental groups received the same anesthetic regime (Hayton et al. 1999). In humans, SSEPs are known to be less sensitive to the injectable anesthetics that we used, than to inhalation anesthetics (Becker et al., 2017).

(c) The lumbar spinal cord apoptosis observed in severe HI models (20 min of submersion) may not necessarily be the postnatal hallmark of the moderate HI model (17 min of submersion) used in this experiment [22].

(d) Evidence supports more neurological impairments and higher mortality for male preterm infants. In the line with the human data, behavioral studies have consistently shown that male sex is associated with an increased risk of long-term motor deficits in the rat preterm HIE [10]. Therefore, only male rats were included in this study.

And

Our study found that the perinatal HI insult did not influence the SSEPs. A previous study with SSEPs after neonatal stroke in the rat, found profound unilateral changes in SSEPs one week after the insult, but recovery of the deficits 3 weeks postnatally. This may indicate large-scale plasticity of the somatosensory networks even after a unilateral neonatal injury [21]. It is thus possible that sufficient recovery may occur 11 months after the global asphyctic insult and ameliorate changes observed in evoked potentials.

Reviewer 1: 4. The authors have to show data from SSEP experiments immediately after HI and that can be a selling point for the manuscript.

Reply: The value of our manuscript is that we provide long-term data on SSEPs in rats. In reference 23, the authors studied SSEPs at 1 and 3 weeks after HI in a different rat model. In our case, we did use SSEP as a diagnostic procedure and not as a prognostic tool in a terminal experiment after long-term behavioral follow-up.

Reviewer 2 Report

This study provides vital results for those interested in EPs for HIE/CP

More discussion of the changes in behaviour are required for the two groups - was there any motor development difference detected visually? 

Were the HIE pups mixed with controls in adopted families?

Would it be possible to more closely match the NICU environment a HIE infant might find itself in?

In limitations it should be added that rodent models of HIE/CP have been found to be very difficult (high mortality and high care needs eg bladder control). Larger animal studies may be needed.

Author Response

We would like to thank you for your time and the helpful feedback. We have proceeded to make a R1 version of our document making changes as suggested by the reviewers. Cut downs will not appear in the new R1 version. However, adjusted new text has been inserted as colored RED TEXT.

We thank the reviewers for their input and hope this revised version (R1) be accepted for publication. Of course, we look forward to further discussions on this manuscript.

Reviewer 2: 1. This study provides vital results for those interested in EPs for HIE/CP

More discussion of the changes in behaviour are required for the two groups - was there any motor development difference detected visually? 

Reply: We reported the behavioral outcomes of these rats in an earlier paper cited in the current work (reference 14). We changed the following wording to make it clear that this cohort is the same rats as in our earlier work.

Lines 62-71. In this respect, our experimental group is a cohort of male adult rats that recovered from moderate global perinatal HI, that has shown cognitive (memory) as well as motor abnormalities (hyperactivity) within the spectrum of ADHD at the age of 6-8 months [14]. To test whether this cohort has a problem in coordination (DCD), we evaluated the sensory functioning using a bilateral tactile stimulation test and we recorded cortical SSEPs after unilateral tibial nerve stimulation at 11 months of age [14].

The present study reports the outcomes of the SSEPs and behavioral tasks of sensory functioning of male adult rats subjected to moderate perinatal HI. We hypothesized that long-term sensory deficits were relevant to the combined sensory and motor deficits previously demonstrated in this cohort of male rats after perinatal HI.

Reviewer 2: 2. Were the HIE pups mixed with controls in adopted families?

Reply: Yes. We clarified this in the methods section (lines 85-89):

The control Cesarean-section rats (C-section) were delivered from the same litters without submersion and placed in the closed pediatric incubator to recover with their HI littermates. After recovery, the mixed litter of HI and C-section pups were placed with a foster mother that had given birth the day before to minimize the impact of maternal care differences on outcomes.

Reviewer 2:3. Would it be possible to more closely match the NICU environment a HIE infant might find itself in?

Reply: We have tried to match the NICU environment as closely as possible, with a global submersion model which models the global nature of birth asphyxia instead of the Rice-Vannuci model of unilateral carotid artery occlusion followed by hypoxia. Pups were left to recover in a human-grade closed pediatric incubator similar to the ones used in the NICU.  And then they were placed with a foster mother. The nest got disturbed daily for remarking so we could distinguish our experimental groups, which would model frequent medical intervention seen in the NICU. We would love additional suggestions on how we can further refine our model but we need a foster mother to nurse the baby rats so it would not be possible to entirely model the NICU environment.

Reviewer 2:4. In limitations it should be added that rodent models of HIE/CP have been found to be very difficult (high mortality and high care needs eg bladder control). Larger animal studies may be needed.

Reply: Whilst our model has a relatively high mortality of 39%, as we stated in the methods section, the global asphyxia model we use produces animals with subtle behavioral deficits during functional tests such as the open field test for locomotion and the object recognition test for memory (reported in reference 14). This model does not produce gross disabilities like impaired bladder control or gross motor deficits which can be observed without behavioral testing. It is reflective of the majority of HI survivors, which have difficulties at school, without overt disability. We added a sentence about large animal models to our limitations section (lines 218-222):

(e) Our global HI insult was conducted around the time of birth, when the rodent brain development is comparable to human infants born at a very low gestational age [10]. Studying SSEPs in larger animal models with brain development comparable to humans, such as sheep, may be more predictive of the human clinical situation [24]. However, larger animals are less suitable for chronic, long term studies.

Round 2

Reviewer 1 Report

Dear authors, 

Thank you for making the required changes to the manuscript. The additional added information provide a clearer understanding of the manuscript.